# Zebrafish *Oxr1a* Knockout Reveals Its Role in Regulating Antioxidant Defenses and Aging

**DOI:** 10.3390/genes11101118

**Published:** 2020-09-24

**Authors:** Hao Xu, Yu Jiang, Sheng Li, Lang Xie, Yi-Xi Tao, Yun Li

**Affiliations:** 1Institute of Three Gorges Ecological Fisheries of Chongqing, College of Fisheries, Southwest University, Chongqing 400715, China; xuhao@email.swu.edu.cn (H.X.); memcpy@email.swu.edu.cn (L.X.); tyx0914@email.swu.edu.cn (Y.-X.T.); 2Key Laboratory of Freshwater Fish Reproduction and Development (Ministry of Education), Key Laboratory of Aquatic Science of Chongqing, Southwest University, Chongqing 400715, China; jjhehe@email.swu.edu.cn (Y.J.); leeseung@email.swu.edu.cn (S.L.)

**Keywords:** *Oxr1a*, antioxidant defenses, oxidative stress, aging, p53 signaling pathway, apoptosis, zebrafish, vertebrates

## Abstract

Oxidation resistance gene 1 (*OXR1*) is essential for protection against oxidative stress in mammals, but its functions in non-mammalian vertebrates, especially in fish, remain uncertain. Here, we created a homozygous *oxr1a*-knockout zebrafish via the CRISPR/Cas9 (Clustered Regularly Interspaced Short Palindromic Repeats/CRISPR associated protein 9) system. Compared with wild-type (WT) zebrafish, *oxr1a^−/−^* mutants exhibited higher mortality and more apoptotic cells under oxidative stress, and multiple antioxidant genes (i.e., *gpx1b*, *gpx4a*, *gpx7* and *sod3a*) involved in detoxifying cellular reactive oxygen species were downregulated significantly. Based on these observations, we conducted a comparative transcriptome analysis of early oxidative stress response. The results show that *oxr1a* mutation caused more extensive changes in transcriptional networks compared to WT zebrafish, and several stress response and pro-inflammatory pathways in *oxr1a^−/−^* mutant zebrafish were strongly induced. More importantly, we only observed the activation of the p53 signaling and apoptosis pathway in *oxr1a^−/−^* mutant zebrafish, revealing an important role of *oxr1a* in regulating apoptosis via the p53 signaling pathway. Additionally, we found that *oxr1a* mutation displayed a shortened lifespan and premature ovarian failure in prolonged observation, which may be caused by the loss of *oxr1a* impaired antioxidant defenses, thereby increasing pro-apoptotic events. Altogether, our findings demonstrate that *oxr1a* is vital for antioxidant defenses and anti-aging in zebrafish.

## 1. Introduction

Reactive oxygen species (ROS) are a byproduct of aerobic metabolism. It has been established that, in eukaryotic cells, mitochondria are an important source of ROS production [1]. Low amounts of ROS have beneficial effects on many essential biological processes such as angiogenesis [2], wound healing [3], the killing of invading pathogens [4] and tissue repair [5]. However, the excessive generation of ROS causes serious problems to bodily homeostasis and leads to an accumulation of oxidative DNA damage and spontaneous mutagenesis [6,7,8], which are associated with diseases including neurodegeneration, cancer and aging [9,10,11].

Oxidation resistance 1 (*OXR1*) is a conserved gene family that only occurs in eukaryotes and is mainly participates in the protection against deleterious ROS. The earliest reports found that human *OXR1* is able to prevent DNA oxidative damage in the DNA-repair-deficient *Escherichia coli* mutants [12], and, when the protein is expressed in the mitochondria of *Saccharomyces cerevisiae* mutants lacking *OXR1*, it is sufficient to suppress oxidative damage [13]. The knockdown of *OXR1* in *Anopheles gambiae* [14] increases its mortality under oxidative stress; a similar result was observed in *Bombyx mori OXR1* mutants [15]. In mice, the genetic ablation of *OXR1* leads to lethal neurodegeneration [16]. As a result, cerebellar ataxia and death before Postnatal Day 26 (P26). In a mice model of amyotrophic lateral sclerosis (SOD1^G93A^), overexpression of *OXR1* has been shown to prolong the lifespan of mice by improving motor deficits and delaying the onset of symptoms [17]. In conclusion, the antioxidant properties of *OXR1* have been confirmed in multiple animal and cell models. Moreover, the roles of *OXR1* also include mitochondrial morphology maintenance [13,18,19], innate immune defense [20,21,22] and the regulation of aging [23,24] and the cell cycle [25]. However, despite all this, its function in the largest vertebrate group—fish—has not been reported. 

Fish, as with mammalian species, also have a relatively complete antioxidant system that maintain cellular ROS at safe levels and avoid oxidative damage. These antioxidants mainly include glutathione peroxidase (GPx), catalase (CAT), superoxide dismutase (SOD), glutathione reductase (GR) and glutathione S-transferase (GST). It has been documented that *OXR1* is able to elevate the expression of these antioxidant actors to reduce oxidative damage in aerobic organisms [19]. Nevertheless, there are still no research data to support the antioxidant functions of *OXR1* in fish. At present, the study of *OXR1* in fish has been limited to gene expression and protein activity analysis. In zebrafish, *oxr1a* (homolog of human *OXR1*) is highly expressed maternally and zygotically at the earliest stages of embryonic development. In the larval stage, it is detected in the central nervous system of brain [26]. Similar to previous reports [16,24], zebrafish OXR1A protein is unable to directly decompose ROS [27]; thus, *oxr1a* has been considered to be a sensor of cellular oxidative stress and to indirectly participate in the scavenging of ROS. Except for the characterization of gene expression and protein activity, there is no information on the study of *OXR1* in zebrafish. Thus, the current research works alone do not seem to allow us to fully understand the roles of *OXR1* in fish. 

In this work, we studied in detail the spatiotemporal expression patterns of *oxr1a* in zebrafish early developmental stage and adult tissues. We also employed CRISPR/Cas9 gene editing technology to establish a null allele of *oxr1a* (homolog of human *OXR1*) in zebrafish to elucidate its molecular functions in vivo. Our data confirm that zebrafish with *oxr1a* deficiency displayed a shortened lifespan, reduced fertility and decreased resistance to oxidative stress. Furthermore, we also examined the role of *oxr1a* in genome-wide transcription regulation by transcriptome sequencing (RNA-seq), revealing the importance of *oxr1a* for balancing transcriptional networks, regulating the early oxidative stress response and modulating apoptosis. 

## 2. Materials and Methods

### 2.1. Animal Maintenance

In this study, wild-type (WT; AB line) and *oxr1a* knockout zebrafish were used as experimental materials. Zebrafish were cultured in a flow-through aquarium, including biological filter tank, UV sterilization, 28 °C constant temperature and 14/10 h light/dark cycle. The larvae and adult fish were fed twice daily with initial diets and brine shrimp (Zeigler, Gardners, PA, USA), respectively. All animal experiments were conducted in accordance with the Guiding Principles for the Care and Use of Laboratory Animals and were approved by the Committee for Laboratory Animal Experimentation at Southwest University, China (IACUC Issue No. 2018022611; Approval date: 26 February 2018).

### 2.2. Whole-Mount in Situ Hybridization (WISH)

Zebrafish embryos or larvae were collected at different developmental stages according to the experimental requirements and immediately fixed overnight at 4 °C with 4% paraformaldehyde (PFA) in phosphate-buffered saline (PBS), and then dehydrated with 100% methanol and stored at −20 °C. Whole-mount in situ hybridization (WISH) for *oxr1a*, *gstp1*, *gstp2* and *prdx1* genes was performed as reported previously [28], and the gene-specific primers for the synthesis of dig-labeled RNA probes are listed in Appendix A. Photographs were taken with a LEICA DM3000 microscope (Leica, Wetzlar, Germany). 

### 2.3. Establishment of Zebrafish Oxr1a Mutant Line

Zebrafish *oxr1a* gene knockout was performed using a CRISPR/Cas9 system, as previously described [29]. Briefly, the gRNA (0.1–0.2 ng) and Cas9 mRNA (0.3–0.5 ng) were mixed and then injected into the one-cell stage of WT embryos. At 72 h post fertilization (hpf), we isolated 6–10 pooled embryos with normal morphology and used a pair of specific primers (*oxr1a*-F1/R1) for the PCR amplification of genomic DNA. The PCR fragments spanned the targeting site of *oxr1a* gene knockout; thus, the knockout efficiency in F0 generation could be determined by Sanger sequencing. Next, we cultivated the F0 mutants to adulthood, and then outcrossed with WT zebrafish to obtain F1 offspring. Subsequently, the genotypes of heterozygous F1 were identified by sequencing analysis of the genomic DNA isolated by tail fin clipping, and then males and females carrying the same mutation types were collected for further obtaining homozygous F2 mutants. In addition to DNA sequencing, we also performed the heteroduplex motility assays to obtain individuals with different genotypes as previously reported [30]. 

### 2.4. Determination of Total Lipid Content

According to the Folch method [31], we determined the total lipid content of whole zebrafish. Briefly, individual zebrafish were exposed to MS-222 and weighed, and then samples were homogenized thoroughly in 500 μL mixture of chloroform: methanol (2:1). Next, add 100 μL 0.9% NaCl solution to the homogenates, and then vortexed and centrifuged at 3000 rpm for 30 min. The lower layer was collected, dried and weighed. This weight indicates the total lipid content per fish. 

### 2.5. Hydrogen Peroxide (H_2_O_2_) Treatment and Apoptosis Assay

To assess the impact of zebrafish *oxr1a* knockout on the resistance to oxidative stress, 5 days post fertilization (dpf) zebrafish larvae were exposed to an exogenous ROS, H_2_O_2_. According to previous reports [32,33] and preliminary experiments, we selected 2 mM H_2_O_2_ as the inducing dose of oxidative stress. Mortality was monitored in each 3-h interval. The death rates were compared between WT and *oxr1^−/−^* mutant zebrafish at 3, 6, 9 and 12 h after exposure to H_2_O_2_. For apoptosis assay, the in situ cell death detection kit (Beyotime, Shanghai, China) was used in our work, as described previously [34]. Briefly, larvae were collected and fixed with 4% PFA overnight. Next, larvae were washed with PBS at least three times and then treated with 1% protease K (without DNase) for 30 min. Subsequently, larvae were rinsed with PBS three times again, and then were labeled with terminal deoxynucleotidyl transferase reaction mixture for 60 min at 37 °C under dark conditions. Finally, samples were washed again with PBS, and then photographs were taken using a LEICA DM6000 microscope (Leica, Wetzlar, Germany).

### 2.6. Histological Analysis and Fertility Assessment

After MS-222 anesthesia, adult zebrafish were sacrificed to dissect the intact gonadal tissues from WT and *oxr1a^−/−^* mutant zebrafish. The gonadosomatic index (GSI) were calculated according to gonad weight/body weight × 100%. Next, we fixed the gonads with 4% PFA and embedded them in paraffin for tissue section (6 μm) and then stained with hematoxylin and eosin (H&E). The images were taken with a LEICA DM 3000 microscope (Leica, Wetzlar, Germany). 

As described in a previous report [35], 12 pairs of adult zebrafish include WT and *oxr1a^−/−^* mutant zebrafish were one-to-one paired and mated in each 7 day interval as follows: WT male × WT female; WT male × *oxr1a^−/−^* female; *oxr1a^−/−^* male × WT female; and *oxr1a^−/−^* male × *oxr1a^−/−^* female. The numbers of spawned individuals were recorded for each batch. The ovulation rate = number of spawned individuals/total number of individuals × 100%.

### 2.7. RNA Sequencing and Real-Time qPCR Analysis

For the extraction of total RNA, TRIZOL Reagent (Invitrogen, Carlsbad, CA, USA) was applied in our study. Sixty larvae were pooled together to obtain one RNA sample. For each sample, we employed SmartSpec^TM^ Plus spectrophotometer (Bio-Rad, Hercules, CA, USA) to measure the total RNA content, and the RNA integrity was assessed by agarose gel electrophoresis. In our research, we pooled triplicate RNA samples from each biological group to minimize the total number of RNA samples used for sequencing. Library construction, sequencing and the main bioinformatics analysis were conducted by the Beijing Genomics Institute (BGI), China. For library construction, RNA quality and integrity were assessed with an Agilent 2100 BioAnalyzer (Agilent Technologies, Palo Alto, CA, USA). Each sample must meet the conditions of 28S/18S RNA ratio ≥ 1.8 and RNA integrity number (RIN) > 8.5 for further processing. In short, the mRNA was isolated from total RNA using oligo (dT) beads, and then fragmented and reverse transcribed into the first strand of cDNA with random primers. Next, the DNA polymerase I and RNase H were used to synthesis the second strand of cDNA and purified them with a PCR extraction kit (QiAquick, Hilden, Germany). The fragments were then subjected to end repair and dA-tailing and ligated to Illumina adapters. The size of ligation products was separated by agarose gel electrophoresis, and the fragments were collected for PCR amplification and then sequenced using a HiSeq2000 instrument (Illumina, San Diego, CA, USA). After RNA sequencing, the clean reads were mapped to the genome sequence of zebrafish (GRCz11) and reference genes (National Center for Biotechnology Information (NCBI) sequence database RefSeq) using SOA Paligner/SOA2 [36]. In this study, FPKM (fragments per kilobase of transcript per million mapped reads) was used to calculate the abundance of gene transcripts, and differential expression levels with a more than two folds change were considered as differentially expressed. The sequencing data were deposited in the NCBI Sequence Read Archive under the accession number PRJNA649399.

For qPCR analysis, the CFX Connect™ Real-Time PCR Detection System (Bio-Rad, San Diego, CA, USA) were employed in our work. Three biological replicates were performed, and each sample was measured in triplicate. For each RNA sample, about 2 μg of total RNA were reverse transcribed to cDNA using oligo (dT) as a primer. The gene specific primers were designed with Primer Premier 5.0 software. In this study, eukaryotic translation elongation factor 1 α 1 (*eef1a1l1*) was used as an internal reference for gene expression normalization. All primers are listed in Appendix A. The amplification was conducted in 20 μL of solution containing 2 × SYBR Green PCR Master Mix (10 μL), cDNA templates (1 μL), each primer (0.4 μL) and nuclease-free water (8.2 μL). The procedure of qPCR amplification was conducted: 95 °C for 3 min, followed by 40 cycles of 95 °C for 10 s and 60 °C for 60 s. The melting curve of the PCR products were measured from 65 to 95 °C, and the purity of the reaction product was determined by the observation of a single melt peak. The expression levels of the tested reference genes were determined by CT values and calculated by 2^−△△^Ct.

### 2.8. Statistical Analysis

Data are expressed as means ± SE. Differences among groups were analyzed by one-way ANOVA with a Tukey’s post-hoc test was performed. For all experiments, *p* < 0.05 was considered statistically significant.

## 3. Results

### 3.1. Zebrafish oxr1a Gene Structure and Expression Profiles

The zebrafish *oxr1a* gene (Gene ID: 571418) is located on *Chr16* and spans a region of 324.68 kb (Figure 1A). When researching the zebrafish *oxr1a* gene from the National Center for Biotechnology Information (NCBI), we found that the gene encodes at least five transcripts and is generated by the alternative splicing of 17 exons (Figure 1B). These isoforms range from 693 to 804 amino acids in length, and each contains three protein domains: LysM, GRAM (glucosyltransferases, rab-like GTPase activators and myotubularins) and TLDc (TBC, LysM, domain catalytic) (Figure 1C). In the zebrafish chromosome, the *oxr1a* gene and seven neighboring genes (*dcaf13, rims2, dcstamp, dpys, zfmp2, sqle* and *rnf144b*) have orthologs located on human *Chr8* and mouse *Chr15*, respectively (Figure 1D), suggesting that zebrafish *oxr1a* gene has a syntenic relationship and conservative function with mammal *OXR1* gene. In addition, *oxr1b* (Gene ID: 570476), a paralogous gene of *oxr1a*, was found on *Chr19*, and three different transcription isoforms, including *oxr1bx1*, *oxr1bx2* and *oxr1bx3*, are generated by alternatively splicing of 18 different exons. However, in addition to the full-length transcripts, *ox1b* has a unique short transcript that contains only one TLDc domain (Appendix A). Their structural difference implies that the two paralogs may have certain functional differentiation.

Next, we examined *oxr1a* expression in embryos and tissues of WT zebrafish. As shown in Figure 2A, *oxr1a* is highly expressed at the earliest stages of embryos, but sharply declined at the shield stage. It further decreases to 0.029% of the one-cell phase at the 75%-epiboly stage and then gradually increases during subsequent development. Whole-mount in situ hybridization (WISH) further confirmed that *oxr1a* is a maternally expressed gene (Figure 2C) and showed that the gene is ubiquitously expressed from the two-cell to the sphere stage. At 20 h post fertilization (hpf), expression becomes restricted to the somite but then gradually transfers to the head after 30 hpf, including the brain, olfactory bulbs, retina and neurons. In adulthood, the distribution of *oxr1a* was determined in all tissues (Figure 2B), including brain, eye, gill, heart, liver, spleen, muscle, intestine, kidney and gonad, with especially high abundance in the brain, eye and testis. The high expression levels of *oxr1a* in embryos, gonads and the central nervous system suggested it may play important roles in early development, reproduction and antioxidant defenses.

### 3.2. Generation of Oxr1a-Knockout Zebrafish

To define the molecular function of *oxr1a* in zebrafish, we targeted the eighth exon of *oxr1a* using CRISPR/Cas9-mediated genome editing so that all transcripts lost their function (Figure 3A). Seventy-two hours after the injection of gRNA and cas9 mRNA, a high frequency of somatic mutation was identified by sequencing from normally developed F0 embryos, and mutations correctly occurred at the target sites (Appendix A). The remaining F0 embryos were raised to adulthood and genotyped by sequencing in F1 generation. Upon DNA sequencing (Figure 3B) and heteroduplex motility assay in the F2 generation (Figure 3C), it is revealed that an 8 bp (ctgtgagg) was deleted in exon 8 of the *oxr1a* gene, which leads to an open reading frame-shift of all transcripts and resulted in the early termination of protein translation. It can be predicted that the truncated protein lost the GRAM and TLDc domain (Figure 3D). Further RT-PCR and qPCR using a deletion-specific primers (*oxr1a*-F2/R2) also determined that transcription of *oxr1a* is completely interrupted in homozygous mutants (Figure 3E,F). 

### 3.3. Characterization of Zebrafish Oxr1a Mutant Phenotypes

The OXR1 gene has been documented to be associated with longevity, and the oxidative stress produced by *OXR1* mutation accelerates animal aging [16,24,37]. Therefore, we gave priority to the investigation of the impact of *oxr1a* mutation on zebrafish lifespan. In this study, we obtained WT and *oxr1a^−/−^* mutant zebrafish by crossing heterozygous mutants. Under standard laboratory conditions, we compared the survival duration between WT and *oxr1a^−/−^* mutant zebrafish from 2 to 18 months post-fertilization. As shown in Figure 4A, the survival of *oxr1a^−/−^* mutants was indistinguishable with WT zebrafish before eight months of age. However, after 10 months of age, we found that the death rate of *oxr1a^−/−-^* mutant zebrafish increased rapidly, and none of the mutant individuals survived more than 16 months; however, WT zebrafish exhibited higher survival rates at the corresponding time. Obviously, the lifespan of *oxr1a^−/−^* mutant zebrafish is significantly lower than that of the WT zebrafish. Meanwhile, we also observed that the morphology of *oxr1a^−/−^* mutant zebrafish is extremely emaciated at 14 months of age (Figure 4B). Compared with WT zebrafish, although the standard length is only slightly different (Figure 4C), the body weight (Figure 4D) and total lipid content (Figure 4E) of *oxr1a^−/−^* mutants are distinctly reduced by 45.21% and 27.61%, respectively. Thus, it appears that *oxr1a^−/−^* mutants becomes increasingly thin with increasing age. 

In addition, we also found that the fertility of *oxr1a^−/−^* mutant females was severely impaired. As shown in Figure 5A and Appendix A, after continuous multiple fertilization, most of the mutant females were unable to spawn successfully at five months of age, whereas the age-matched WT females and mutant males still were fertile. Through histological analyses, we only in *oxr1a^−/−^* mutant females observed several early developmental stages of oocytes, including the primary growth stage (PG), previtellogenic stage (PV) and early vitellogenic stage (EV) (Figure 5D), and the GSI was markedly reduced in *oxr1a^−/−^* mutant females (9.2%) compared to WT females (16.2%) (Figure 5B). No evident difference was observed in terms of the morphology and GSI of the testis between WT and *oxr1a^−/−^* mutant males (Figure 5C,E). These results suggest that *oxr1a* mutation may result in premature ovarian failure in female zebrafish.

### 3.4. Oxr1a Mutant Zebrafish are Hypersensitive to Oxidative Stress

According to previous reports [24,37], it is speculated that the aging and premature ovarian failure in *oxr1a^−/−^* mutants may be induced by *oxr1a* mutation impairing antioxidant defenses, thereby increasing oxidative damage events. Thus, we examined the sensitivity of *oxr1a^−/−^* mutants toward oxidative stress. To assess the impact of zebrafish *oxr1a* knockout on the ROS sensitivity, we treated 5 dpf larvae with H_2_O_2_ to induce oxidative stress and recorded their death in detail. As expected, the death rates of *oxr1a^−/−^* mutants were significantly higher than that of the WT larvae during 2 mM H_2_O_2_ stress (Figure 6A). After exposure to H_2_O_2_ for 6 h, approximately 50% of *oxr1a^−/−^* mutant larvae died, whereas the death rates in WT zebrafish were only around 15%. After being consistently exposed for 12 h, the death rates of *oxr1a^−/−^* mutant zebrafish reached 95%, whereas only 70% of WT zebrafish died under the same treatment. 

By qPCR, we detected several antioxidant genes include *gpx1b, gpx4a*, *gpx7* and *sod3a* in *oxr1a^−/−^* mutant zebrafish which were significantly downregulated compared to WT zebrafish (Figure 6B), indicating that *oxr1a* mutation impairs the ability of zebrafish to detoxify ROS, thereby possibly inducing apoptosis during oxidative stress. Thus, we further measured the apoptosis during H_2_O_2_ stress by TUNEL assay. As shown in Appendix A, almost no apoptosis was observed in both WT and *oxr1a^−/−^* mutant zebrafish before H_2_O_2_ stress. However, after exposure to H_2_O_2_ for 3 and 6 h, a high aggregation of apoptotic cells was observed in olfactory bulbs of *oxr1a^−/−^* mutant larvae, which is a key part of the neural system responsible for the sense of smell. In contrast, only a few TUNEL-positive cells were found in the olfactory bulbs of WT zebrafish (Figure 6C).

The kelch-like ECH-associated protein 1–NF-E2-related factor 2 (Keap1–Nrf2) system is regard as a master regulator of antioxidant defenses in vertebrates [32,38]; we therefore measured the effect of *oxr1a* mutation on the Keap1/Nrf2 signaling pathway. Unexpectedly, no differences were observed regarding the expression of genes including *keap1a*, *keap1b*, *nfe2l2a* and *nfe2l2b* and their target genes *gstp1*, *gstp2* and *prdx1* between WT and *oxr1a^−/−^* mutant zebrafish (Figure 7A,B).

### 3.5. Transcriptome Analysis of oxr1a^−/−^ Mutant Zebrafish by RNA Sequencing

The *oxr1a* mutation resulted in zebrafish larvae being more sensitive to oxidative stress, as demonstrated by the comparison of the resistance to H_2_O_2_ stress and apoptosis and the differential expression of antioxidant genes between WT and *oxr1a^−/−^* mutant zebrafish. For a more comprehensive analysis of the role of *oxr1a* on zebrafish for protection against oxidative stress, RNA-Seq was applied to compare the early oxidative stress response between WT and *oxr1a^−/−^* mutant zebrafish. Samples (WT_ctrl and *oxr1a^−/−^*_ctrl) were collected before H_2_O_2_ stress, while samples (WT_H_2_O_2_ and *oxr1a^−/−^*_H_2_O_2_) were obtained after larvae were exposed to H_2_O_2_ for 3 h. In this work, after the quality filtering, RNA-seq generated more than 21 million clean reads for each sample, and more than 88% clean reads could be mapped to the zebrafish reference genome (GRCz11), of which more than 62% of the reads displayed unique matches. Furthermore, more than 75% of clean reads could be mapped to reference genes, of which more than 66% of the reads showed completed matching. In addition, more than 15,000 genes exhibited more than 50% sequence coverage (Table 1). 

In the present study, the criterion of more than two-fold change in expression was used to determine differentially expressed genes (DEGs). The results show that the expression of 304 genes in WT zebrafish displayed a significant change under H_2_O_2_ stress, of which 213 genes were upregulated and 91 genes were downregulated (Figure 8A and Appendix A). In *oxr1a^−/−^* mutant zebrafish, the amounts of differentially expressed genes (DEGs) increased to 550, including 411 upregulated genes and 139 downregulated genes (Figure 8A and Appendix A). Interestingly, among these DEGs, 269 upregulated genes (55.81%) and 115 downregulated genes (55.83%) were generated only in *oxr1a^−/−^* mutant zebrafish after H_2_O_2_ stress (Figure 8B), suggesting that *oxr1a* mutation significantly affects the transcriptional regulations of zebrafish during oxidative stress. 

### 3.6. Oxr1a Mutation Increases Early Stress Response Levels during Oxidative Stress

Oxidative stress causes extensive changes in the transcriptional network of mammalian cells, and a large amount of stress response genes were differentially expressed [39,40]. Generally, the fold changes in the expression of these genes also reflect the stress response levels of the cells [25]. In our work, 166 DEGs (142 up and 24 down) exhibited similar change trends in both WT and *oxr1a^−/−^* mutant zebrafish under H_2_O_2_ stress by Venn analysis (Figure 8B). However, further analysis found that most DEGs displayed a significantly stronger fold change in *oxr1a^−/−^* mutant zebrafish as compared to WT zebrafish (Figure 9A,B). Notably, many of these DEGs are well-known stress response genes, including *dusp1* (dual specificity phosphatase 1)*, dusp5*, *egr1* (early growth response 2a), *egr2a*, *egr3*, *egr4*, *mmp9* (matrix metallopeptidase 9), *mmp13a*, *hsp70l* (heat shock cognate 70-kd protein, like), *hsp70.1*, *hspa4a*, *fosaa* (V-fos FBJ murine osteosarcoma viral oncogene homolog Aa), *fosab*, *fosb*, *fosl1a*, *fosl2*, *junba* (JunB proto-oncogene, AP-1 transcription factor subunit a) and *junbb* (Figure 9C). Furthermore, we also found the fold changes in the expression of numerous inflammatory factors such as *il1b* (interleukin, β), *il8*, *il11a*, *c7b* (complement component 7b), *cxc18b* (chemokine C-X-C motif ligand 18b) and *tnfaip2b* (tumor necrosis factor, α-induced protein 2b) in *oxr1a^−/−^* mutants were greater than that of WT zebrafish (Figure 9C). A previous report has shown that the proteins encoded by these early stress response genes can form a huge regulatory network with more other proteins [25]. Thus, the increased level of early stress response genes in *oxr1a^−/−^* mutant zebrafish indicates the importance of *oxr1a* in regulating the early stress response during oxidative stress. 

To further reveal the role of *oxr1a* in early oxidative stress response, we performed Gene Ontology (GO) enrichment analysis of DEGs in WT and *oxr1a^−/−^* mutant zebrafish, respectively. In *oxr1a^−/−^* mutant zebrafish, the top 20 biological processes include response to stress, defense response, inflammatory response, response to external biotic stimulus, response to other organism, response to biotic stimulus, response to bacterium and response to heat (Figure 9D). However, these GO terms were not observed in WT zebrafish except for response to heat (Appendix A). Obviously, this observation further showed that *oxr1a* plays an important role in regulating the early oxidative stress response.

### 3.7. Oxr1a Mutation Induces the p53 Signaling and Apoptosis Pathway during Oxidative Stress

The KEGG enrichment analysis showed that H_2_O_2_ stress resulted in 27 pathways being considerably affected in *oxr1a^−/−^* mutant zebrafish, whereas only 19 pathways were affected in WT zebrafish (Appendix A). Among these pathways, 13 pathways—ferroptosis, IL-17 signaling, metabolism of xenobiotics by cytochrome P450, NOD-like receptor signaling pathway, TNF signaling, mineral absorption, chemical carcinogenesis, small cell lung cancer, necroptosis, estrogen signaling, toxoplasmosis, osteoclast differentiation and drug metabolism-cytochrome P450—are regulated in both WT and *oxr1a^−/−^* mutant zebrafish. Interestingly, however, we only observed the activation of the p53 signaling pathway and apoptosis pathway in *oxr1a^−/−^* mutants, but not in WT zebrafish (Appendix A). It is well-known that p53 signaling pathway-mediated apoptosis is one of the major signaling pathways involved in the stimulation of oxidative stress [41]. 

According to the RNA-seq data, 12 genes of the p53 signaling pathway were differently expressed in *oxr1a^−/−^* mutant zebrafish after H_2_O_2_ stress (Figure 10). These genes include *cdkn1a* (cyclin-dependent kinase inhibitor), *fas* (fas cell surface death receptor), *gadd45aa* (growth arrest and DNA-damage-inducible), *gadd45ba*, *gadd45bb*, *casp3b* (caspase 3b), *casp8l2*, *LOC100535166* (*thrombospondin-1-like*), *pmaip1* (phorbol-12-myristate-13-acetate-induced protein), *serpine1* (serpin peptidase inhibitor), *serpinel3* and *sesn2* (*sestrin 2*), and all the genes were significantly upregulated. In contrast, only three genes (*serpine1*, *gadd45bb* and *sesn2*) were upregulated in WT zebrafish (Appendix A), while the fold change of upregulation was lower compared with *oxr1a^−/−^* mutants. 

In the apoptosis pathway, 18 genes were differentially expressed in *oxr1a^−/−^* mutant zebrafish after H_2_O_2_ treatment (Figure 11), of which six genes are shared with WT zebrafish: *aifm4* (apoptosis inducing factor mitochondria associated 4), *fosab*, *gadd45bb, pik3r3a* (phosphoinositide-3-kinase), *zgc:162184* and *LOC101886621* (endonuclease G, mitochondrial-like) (Appendix A). A similar observation was found for the p53 signaling pathway: most genes displayed a dramatically stronger upregulation or downregulation in *oxr1a^−/−^* mutant zebrafish (Appendix A). In addition, 12 other genes—*fas*, *gadd45aa, gadd45ba, casp3b, casp3l, casp8l2*, *pmaip1*, *ctss2.1* (cathepsin S), *fosaa*, *fosb*, *mcl1b* (myeloid cell leukemia sequence 1b) and *xiap* (X-linked inhibitor of apoptosis)—were only upregulated in *oxr1a^−/−^* mutant zebrafish (Appendix A).

To further determine the role of *oxr1a* in regulating the p53 signaling pathway during oxidative stress, we checked the relative mRNA levels for 12 DEGs in this pathway by qPCR. As shown in Figure 12A,B, the qPCR results of candidate genes are highly consistent with the RNA-seq data. By a linear regression analysis, we found a significant positive correlation in the fold change of the gene expression ratio between the RNA-seq and qPCR data (Figure 12C). In addition, the *oxr1a* gene mRNA level was also examined, and an extremely low expression was observed in *oxr1a^−/−^* mutant zebrafish (Figure 12B). These observations indicate that the RNA-seq data are high quality and reliable. 

Taken together, the activation of the p53 signaling and apoptosis pathway in *oxr1a^−/−^* mutant zebrafish supported the findings of our previous studies (Figure 6) that *oxr1a* mutation induces apoptosis during oxidative stress, showing that *oxr1a* mutation may mediate the p53 signaling pathway to induce apoptosis. These findings suggest that zebrafish *oxr1a* may act as a sensor in cells to regulate the transcriptional networks required for ROS detoxification during oxidative stress (Figure 12D). 

## 4. Discussion

In this work, we successfully created a homozygous *oxr1a*-knockout line in zebrafish via CRISPR/Cas9 gene editing technology. This mutation was a deletion of 8 bp in exon 8 of the *oxr1a* gene, which leads to a frameshift mutation and generated a truncated protein that loses the GRAM and TLDc domain; the latter is well-known to be the core domain for antioxidant defenses of *OXR1* [16,37]. Our data demonstrate that zebrafish with *oxr1a* deficiency displayed a shortened lifespan, reduced fertility and decreased resistance to oxidative stress. As far as we know, this is the first animal model used to study the functions of oxidation resistance 1 in a non-mammalian vertebrate.

Oxidative stress has been implicated in the aging process [42]. It is well documented that mutations in the *OXR1* gene results in increased sensitivity to oxidative stress, and the lifespan of various animal models lacking *OXR1*, including *Caenorhabditis elegans* [24], *Drosophila melanogaster* [37] and mice [16], is significantly reduced, revealing an important role for *OXR1* in regulating the aging process. In our present study, similar to previous reports, *oxr1a^−/−^* mutant zebrafish displayed a shorter lifespan than WT zebrafish. Concurrently, we also observed that *oxr1a^−/−^* mutants became increasingly thin with age, as demonstrated by measuring the standard length, body weight and total lipid content, suggesting that *oxr1a* has an important effect on the aging process in zebrafish. Interestingly, however, *OXR1* mutants are lethal in juvenile mice [16], so a reproductive role has not been found in mammals. In this study, we unexpectedly found that *oxr1a* mutation impaired the fertility of zebrafish; that is, most mutants became infertile after five months of age. Through anatomical observation and histological analysis, no mature oocytes were observed in the ovary of *oxr1a^−/−^* mutants, indicating that *oxr1a* mutation may leads to premature ovarian failure. A previous report has demonstrated that, after the acute inflammatory reaction during ovulation, many free radicals accumulate in the ovary and cause severe DNA damage in ovarian epithelial cells [43]. Nuclear receptor co-activator (*Ncoa7*) has been shown to play a crucial role in the protecting ovarian tissue against the oxidative stress that occurs during ovulation process [44,45]. Notably, we found that *oxr1a* gene has significant homology with *ncoa7* in genomic structure and conserved domains, suggesting that it may has a conservative function for *oxr1a* in antioxidant protection during ovulation; thus, the premature ovarian failure of *oxr1a^−/−^* mutant zebrafish is most likely caused by the loss of *oxr1a* increasing oxidative damage to the ovary. However, to clarify this issue, further explorations are warranted in the future.

To confirm the role of *oxr1a* in the antioxidant defenses of zebrafish, we selected H_2_O_2_ as an oxidant to induce oxidative stress. Compared with WT zebrafish, *oxr1a* mutant zebrafish larvae showed lower resistance to H_2_O_2_ stress, and more apoptotic cells were detected in the olfactory bulbs, demonstrating that the importance of *oxr1a* in protecting against oxidative stress in zebrafish. In *Anopheles gambiae*, the role of *OXR1* in the transcriptional regulation of *gpx1* and *cat* has been revealed [14]. The knockdown of *OXR1* in HeLa cells also showed a significant downregulation in *gpx2* and *ho-1* expression [25]. In our works, although zebrafish *oxr1a* knockout was shown to have no impact on the expression level of *ho-1*, *gpx1* and *cat*, several other antioxidants (i.e., *gpx1b*, *gpx4a*, *gpx7* and *sod3a*) were significantly downregulated, thus supporting the view of previous reports that *OXR1* indirectly eliminates intracellular ROS by upregulating the antioxidant genes. Moreover, we also studied the effects of *oxr1a* mutation on the Keap1/Nrf2 signaling pathway, which is essential for vertebrates to protect against oxidative stress [32,38]. In this pathway, *nfe2l2a*, *nfe2l2b*, *keap1a*, *keap1b* and their target genes (*gstp1*, *gstp2* and *prdx1*) in mRNA levels were examined, but no significant differences were observed in the expression levels between WT and *oxr1a^−/−^* mutant zebrafish, suggesting that *oxr1a* may participate in detoxifying ROS via an Keap1/Nrf2-independent pathway.

To further elucidate the molecular mechanisms underlying *oxr1a* protecting against oxidative stress in zebrafish, we employed high-throughput sequencing to investigate the impact of *oxr1a* on the transcriptional regulation of gene expression under oxidative stress. In the RNA-seq data, more DEGs were produced in *oxr1a^−/−^* mutants compared to WT zebrafish under H_2_O_2_ stress. Among these DEGs, 166 genes were commonly regulated in both WT and *oxr1a^−/−^* mutant zebrafish, yet most DEGs displayed a stronger up- or downregulation in *oxr1a^−/−^* mutant zebrafish. These genes mainly include *AP-1 (activator protein 1)* transcription factors *fosaa, fosab, fosb,*
*fosl1a, fosl2, junba* and *junbb*; signal transduction *dusp1, dusp5, egr1, egr2a* and *egr3*; and inflammation-related factors *il1b*, *il8*, *il11a*, *c7b*, *cxc18b* and *tnfaip2b*, most of which belong to early stress response genes, as has been reported in mammalian cell lines [39,40,46,47]. By GO enrichment analysis, we further revealed that several biological processes associated with early stress response were activated in *oxr1a^−/−^* mutants, but none of these occurred in WT zebrafish. Obviously, *oxr1a* mutation disrupts the balance of transcriptional regulation in early stress response during oxidative stress. 

According to KEGG enrichment analysis, many pathways were significantly affected in both WT and *oxr1a^−/−^* mutant zebrafish during early oxidative stress response. Interestingly, we only observed the activation of the p53 signaling pathway and apoptosis pathway in *oxr1a^−/−^* mutant zebrafish. It is well documented that the p53 signaling pathway is one of the major pathways to induce apoptosis [48]. In this pathway, four pro-apoptotic genes in *oxr1a^−/−^* mutants were strongly induced by H_2_O_2_ stress, including *fas*, *casp8l2*, *casp3b* and *casp3l*. It is known that *fas* gene is a cell surface death receptor and mainly expressed on a variety of cell surfaces as a homotrimer [49]. The FAS protein binds and activates caspase-8, which in turn directly activates caspase-3 and triggers apoptosis [50]. Our data suggest that *oxr1a* may act to resist oxidative stress-induced apoptosis through the ROS–FAS–Caspase axis of the p53 signaling pathway. This observation is similar to a previous report in human *OXR1*-depleted HeLa cells [25]; the difference is that human *OXR1* regulates apoptosis via the ROS–CYTC–Caspase axis of the p53 signaling pathway.

## 5. Conclusions

In summary, our data demonstrate that *oxr1a* is crucial in protecting against oxidative stress in zebrafish. During oxidative stress, *oxr1a* regulates the required transcriptional networks, including the antioxidant system, early stress response and p53 signaling pathway, to detoxify cellular ROS and prevent cells from oxidative DNA damage-induced apoptosis. In addition, a prolonged observation found that the normal expression of *oxr1a* is essential for the regulation of reproduction and aging in zebrafish. It is known that the occurrence of these two physiological processes is accompanied by high ROS levels [43,44]. Therefore, it is speculated that aging and premature ovarian failure in *oxr1a^−/−^* mutant zebrafish may be caused by the loss of *oxr1a* impairing antioxidant defenses, thereby increasing oxidative damage and pro-apoptotic events.

## Figures and Tables

**Figure 1 genes-11-01118-f001:**
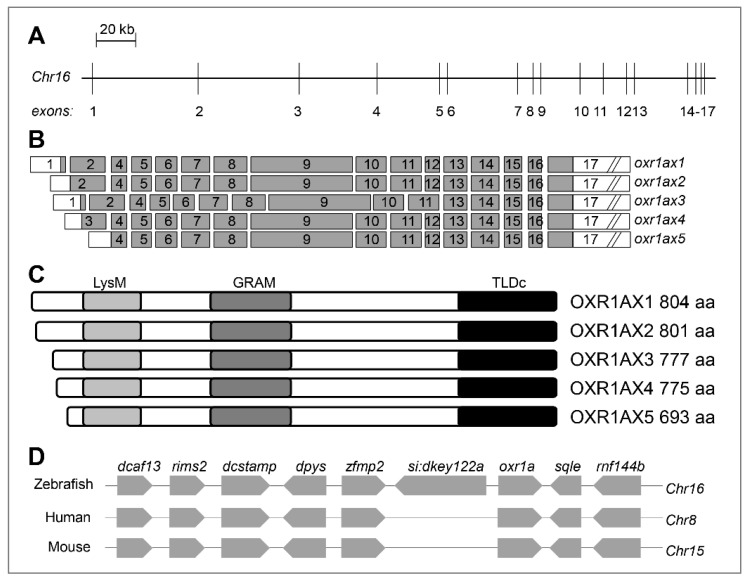
Genomic organization of zebrafish *oxr1a* gene and its synteny analysis. (**A**) Genomic structure of zebrafish *oxr1a* gene. The gene is located on *Chr16* and consists of 17 exons with a span 324,686 bp. The intron length between exons is annotated using a scale. (**B**) Exon composition of *oxr1a* isoforms. The white box indicates an untranslated region and the gray box indicates open reading frame. (**C**) Predicted peptides and conserved domains of *oxr1a* isoforms. All isoforms contain the LysM, GRAM and TLDc domains. The predicted number of amino acids (aa) for each isoform is indicated. (**D**) Synteny analysis: seven adjacent genes (*dcaf13, rims2, dcstamp, dpys, zfmp2, sqle* and *rnf144b*) of zebrafish *oxr1a* gene are in a syntenic relationship with the human and mouse genome oxidation resistance 1 (*OXR1*) gene.

**Figure 2 genes-11-01118-f002:**
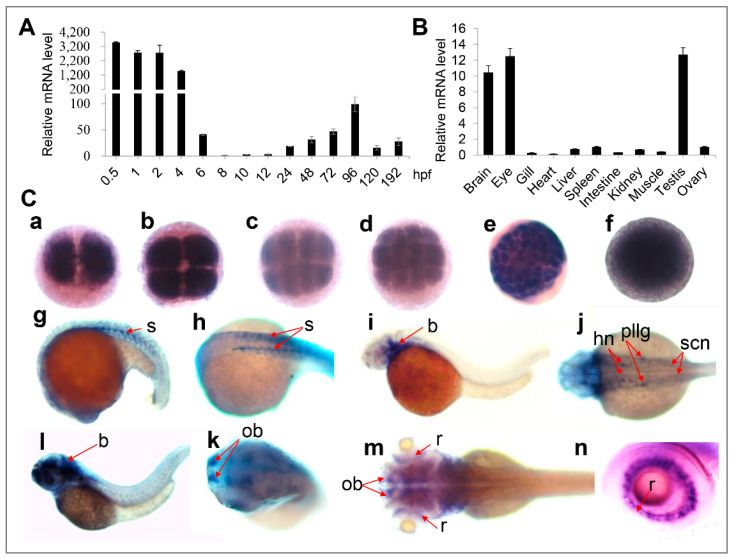
Expression of zebrafish *oxr1a* gene in different embryonic developmental stages and different tissues (**A**) qPCR analysis for the expression profile of *oxr1a* mRNA in wild-type (WT) embryos and larvae from 0.5 to 192 hpf. (**B**) qPCR detected the expression of *oxr1a* in adult tissues at 4 months old. (**C**) Whole mount in situ hybridization analysis (WISH) for the expression profile of *oxr1a* at: (a) two-cells; (b) four-cells; (c) eight-cells; (d) 16-cells; (e) 64-cells; (f) sphere; (g–h) 20 hpf; (i–j) 30 hpf; (l–k) 48 hpf; and (m–n) 72 hpf. The temporal and spatial expression in the somite (s), brain (b), hindbrain nuclei (hn), posterior lateral line ganglion (pllg), olfactory bulbs (ob), spinal cord neuron (scn) and retina (r) are labeled with red arrows. *n* = 20, and the positive rate of expression signal is about 90%.

**Figure 3 genes-11-01118-f003:**
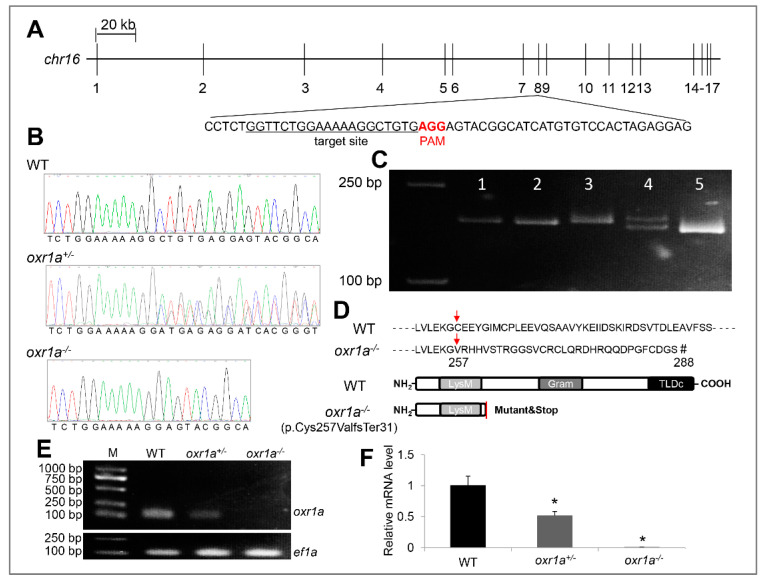
Generation and verification of *oxr1a* knockout zebrafish. (**A**) CRISPR/Cas9 system was employed to edit exon 8 of the *oxr1a* gene. The target site and PAM motif are labeled, respectively. (**B**) DNA sequencing of *oxr1a* gene from WT, *oxr1a^+/-^* and *oxr1a^−/−^* mutant zebrafish. The deletion of 8 bp at the target site was confirmed via Sanger sequencing. (**C**) The genotypes of the F2 animals were identified via the polyacrylamide gel electrophoresis (PAGE) analysis of PCR amplicons. The band size among WT, *oxr1a^+/-^* and *oxr1a^−/−^* mutant zebrafish can be distinguished easily. Lanes 1, 2, and 5 indicate *oxr1a* homozygous mutant; Lane 4 indicates *oxr1a* heterozygous mutant; and Lane 5 indicates WT zebrafish. (**D**) The *oxr1a^−/−^* mutant zebrafish carrying a deletion of 8-bp resulted in the cysteine (**C**) of *oxr1ax1* isoform at position 257 to be mutated to valine (V), and it ultimately generated a predicted truncate OXR1A protein (p.Cys257ValfsTer31) with lacking the GRAM and TLDc domains. Red arrow labels the start site of the amino acids mutation, and the “#” indicates the site of translation termination. Semi-quantitative PCR (**E**) and qPCR (**F**) assay of *oxr1a* mRNA among WT, *oxr1a^+/-^*, and *oxr1a*^−/−^zebrafish using a pair of deletion-specific primers (*oxr1a*-F2/R2). * indicates significant differences (*p* < 0.05).

**Figure 4 genes-11-01118-f004:**
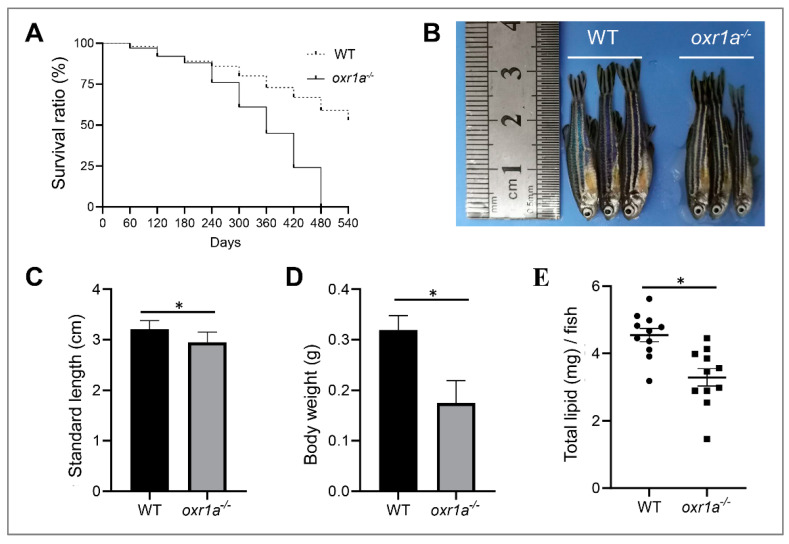
*Oxr1a^−/−^* mutant zebrafish reduced lifespan and impacted physique. (**A**) The survival curves of WT and *oxr1a^−/−^* mutant zebrafish were depicted by Kaplan–Meier assay, *n* = 100 zebrafish in each group. The morphology (**B**); standard length (**C**); and body weight (**D**) were measured in both WT and *oxr1a^−/−^* mutant zebrafish at 14 months old. (**E**) The total lipid content was tested and compared between WT and *oxr1a^−/−^* mutant zebrafish by the Folch method. * indicates significant differences (*p* < 0.05).

**Figure 5 genes-11-01118-f005:**
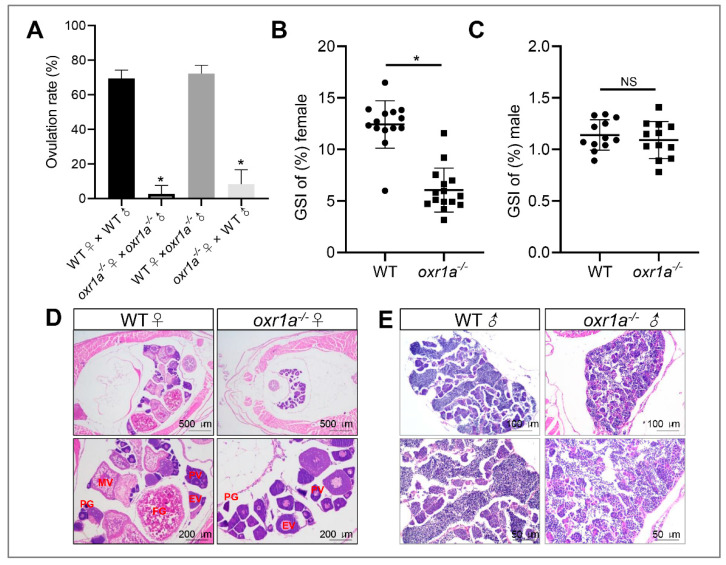
*Oxr1a* mutation impaired the fertility and induced ovarian failure in zebrafish. (**A**) A significant decline of spawning rate in *oxr1a^−/−^* mutant females at five months old. (**B**,**C**) The gonadosomatic index (GSI) scatterplot of WT and *oxr1a^−/−^* mutant zebrafish in both female and male zebrafish, respectively. *n* = 12. (**D**,**E**) H&E staining for histological analysis of WT and *oxr1a^−/−^* mutant zebrafish gonads. PG, PV, EV, MV and FG represent primary growth stage, previtellogenic stage, early vitellogenic stage, midvitellogenic stage and full-grown stage, respectively. * indicates significant differences (*p* < 0.05), while NS indicates no significant difference (*p* > 0.05).

**Figure 6 genes-11-01118-f006:**
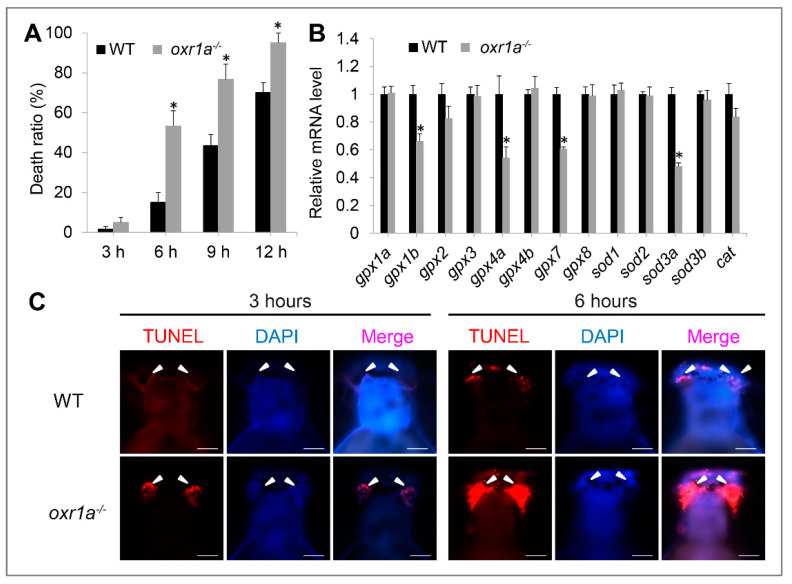
*Oxr1a^−/−^* mutant zebrafish are sensitive to oxidative stress. (**A**) WT and *oxr1a^−/−^* mutant larvae were exposed to 2 mM H_2_O_2_ for 12 h, and the latter showed a higher death rate. (**B**) *Oxr1a* mutation inhibited the expression of antioxidant genes include *gpx1**b*, *gpx4a, gpx7* and *sod3a*. (**C**) Three and 6 h after H_2_O_2_ stress, more apoptotic cells were observed in the olfactory bulbs of *oxr1a^−/−^* mutant larvae by TUNEL staining. The white arrowhead indicates the distribution of TUNEL-positive cells. Bars = 200 μm. * indicates significant differences (*p* < 0.05).

**Figure 7 genes-11-01118-f007:**
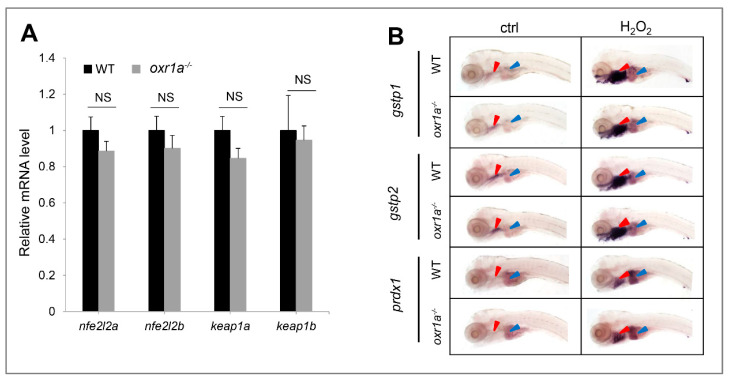
Effect of *oxr1a* mutation on Keap1/Nrf2 pathway during oxidative stress. (**A**) *Oxr1a* mutation does not affect the gene expression of major members in the Keap1/Nrf2 pathway. (**B**) Nrf2 target genes *gstp1*, *gspt2* and *prdx1* in response to H_2_O_2_ for 3 h were examined by WISH. *n* = 20, and the positive rate of expression signal is about 90%; *Oxr1a* mutation does not affect the antioxidant response of Nrf2 target genes. Red and blue arrowheads labeled the gills and liver, respectively. ctrl indicates the larvae were not treated by H_2_O_2_. NS indicates no significant differences (*p* > 0.05).

**Figure 8 genes-11-01118-f008:**
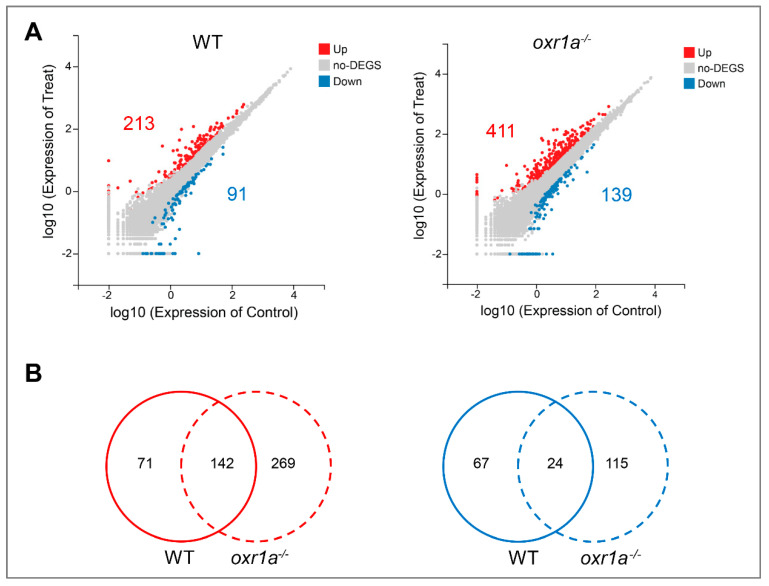
The differential expression analysis in WT and *oxr1a^−/−^* mutant zebrafish during oxidative stress. (**A**) Scatter plot of differential expression genes in WT and *oxr1a^−/−^* mutant zebrafish under H_2_O_2_ stress. The red and blue indicate the number of upregulated and down regulated genes, respectively. (**B**) Venn diagram analysis of commonly or uniquely change for gene expression in WT and *oxr1a^−/−^* mutant zebrafish during H_2_O_2_ stress.

**Figure 9 genes-11-01118-f009:**
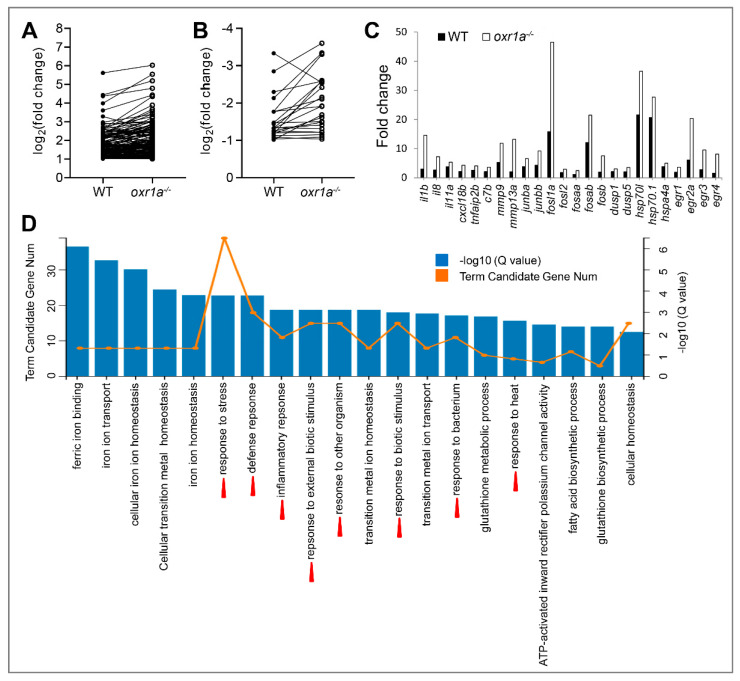
Comparative analysis for differentially expressed genes (DEGs) between WT and *oxr1a^−/−^* mutant zebrafish. (**A**,**B**) Commonly up- or downregulated genes in WT and *oxr1a^−/−^* mutant zebrafish during oxidative stress. Black solid and hollow circles indicate the fold change of gene expression in WT and *oxr1a^−/−^* mutant zebrafish after H_2_O_2_ stress, respectively. (**C**) Most DEGs involved in early oxidative stress response and inflammatory reaction displayed significantly strong changes in *oxr1a^−/−^* mutant zebrafish. (**D**) Gene Ontology (GO) analysis of the DEGs in *oxr1a^−/−^* mutant zebrafish during oxidative stress. Red arrowheads show the early oxidative stress response pathways. The number of DEGs is shown in each biological process.

**Figure 10 genes-11-01118-f010:**
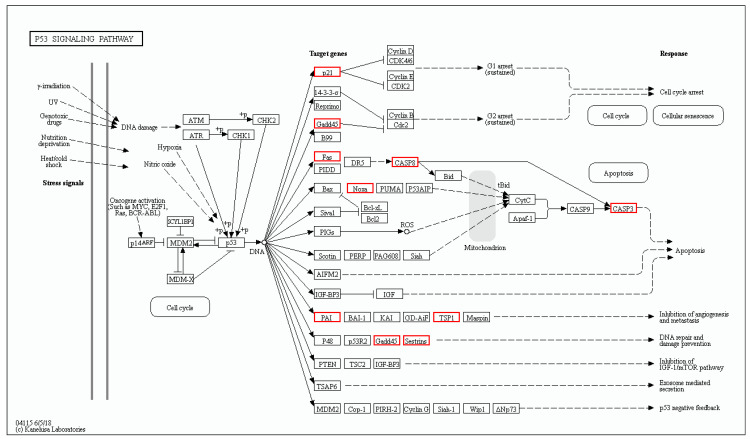
The differentially expressed genes involved in the p53 signaling pathway were annotated by KEGG. The DEGs included *cdkn1a*, *fas, gadd45aa*, *gadd45ba*, *gadd45bb*, *casp3b*, *casp8l2*, *pmaip1*, *LOC100535166*, *serpine1*, *serpinel3* and *sesn2*. Red indicates upregulated genes.

**Figure 11 genes-11-01118-f011:**
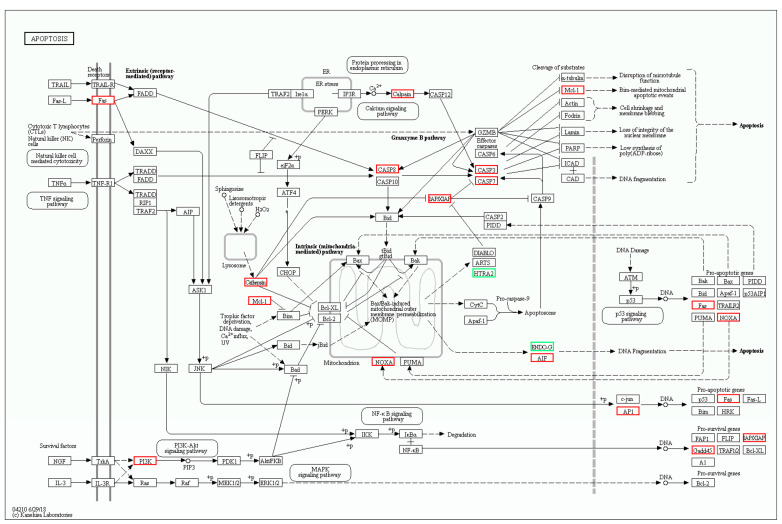
The differentially expressed genes involved in the apoptosis pathway were annotated by KEGG. The DEGs included *fas*, *aiftm4*, *casp3b, casp3l, casp8l2, pmaip1, fosaa*, *fosab*, *pik343a, LOC101886621, ctss2.1, gadd45aa, gadd45ba, gadd45bb, mcl1b, zgc:162184, pik3r3a* and *xiap*. Red and green indicate up- and downregulation of gene expression, respectively.

**Figure 12 genes-11-01118-f012:**
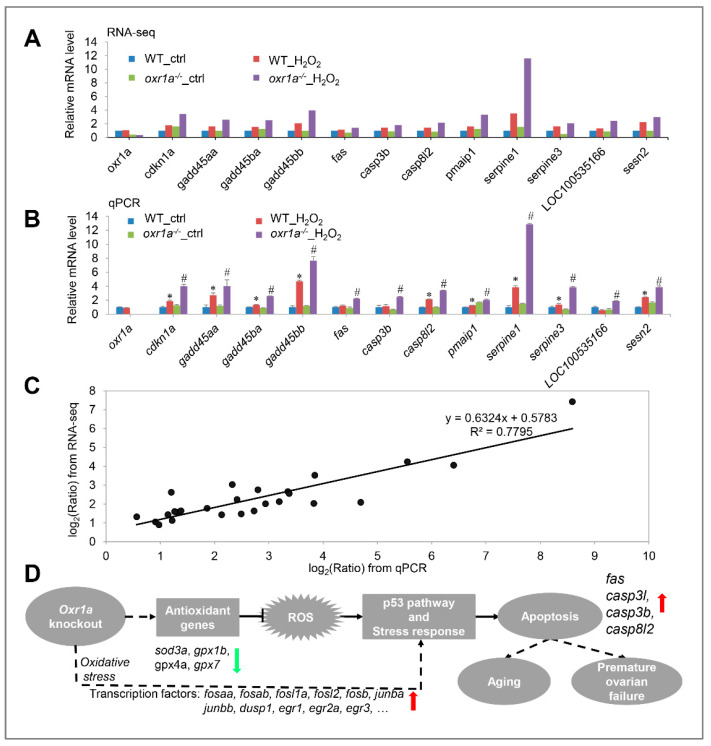
(**A**) RNA-seq data displayed the expression of 12 DEGs in the p53 signaling pathway. (**B**) The same genes were examined by qPCR. (**C**) Scatter plot of gene expression in qPCR (x-axis) and RNA-seq (y-axis). The “Ratio” of log_2_ (Ratio) indicates the fold change in gene expression. The linear relationship is presented by a reference line, and related index *R*^2^ = 0.7795. (**D**) Schematic diagram of *oxr1a* regulating the antioxidant defenses, early stress response and apoptosis. In zebrafish, *oxr1a* scavenges ROS by increasing the expression of several antioxidant genes, thereby modulating early stress response and suppressing apoptosis. Due to the downregulation of multiple antioxidant genes, zebrafish *oxr1a* mutation may enhance DNA damage during oxidative stress, thereby triggering apoptosis via the regulation of *fas*, *casp3b*, *casp3l*, *casp8l2* and several other genes of the p53 signaling pathway. In addition, *oxr1a* may indirectly or directly regulate the early stress response and apoptosis through interaction with transcription factors (e.g., *fosaa, fosab*, *junba, junbb, dsup1* and *egr1*). Red and green arrows indicate up- and downregulation of gene, respectively, in *oxr1a^−/−^* mutant zebrafish during oxidative stress. The solid line indicates a direct interaction, while the dotted line indicates a direct or indirect interaction. * *p* < 0.05 compared to WT_ctrl; # *p* < 0.05 compared to *oxr1a^−/−^*_ctrl.

**Table 1 genes-11-01118-t001:** Summary for the filtering and mapping of reads.

Sample	Total Clean Reads (M)	Reads Mapped to Genome		Reads Mapped to Gene		Mapped Genes at Coverage >50%
		Total Mapping (%)	Uniquely Mapping (%)	Total Mapping (%)	Uniquely Mapping (%)	
WT_ctrl	21.92	88.05	62.87	75.21	66.39	16,185
WT_H_2_O_2_	21.90	88.42	63.32	76.01	67.01	15,935
*oxr1a^−/−^*_ctrl	21.91	88.00	62.94	75.37	66.61	15,906
*oxr1a^−/−^*_H_2_O_2_	21.91	88.14	63.15	75.54	66.68	15,675

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
