# Peer review of "Zebrafish Oxr1a Knockout Reveals Its Role in Regulating Antioxidant Defenses and Aging"

_genes, 2020, doi:10.3390/genes11101118_

Round 1

Reviewer 1 Report

Review of manuscript Genes-921171, titled “Zebrafish Oxr1a knockout reveals its role in regulating antioxidant defenses and aging” by Xu and colleagues:

In this manuscript, the authors have generated knockout zebrafish model of oxr1a by CRISPR/Cas9 and analyzed its effect on fertility and aging of mutant fish. They also performed RNASeq on WT and mutant fish with and without H2O2 treatment to identify the genes and pathways affected by loss of Oxr1 after oxidative stress. As the authors state that he function of OXR1 has been well defined in human, mice and other organisms and thus the study is not novel. However, it adds to our understanding of conserved gene function between zebrafish and mammals and warrants publication as the first study to describe a zebrafish model of oxr1a knockout. The manuscript is well-written but needs careful revision for grammar. This reviewer has the following specific comments.

Major concerns:

  1. The authors should follow zebrafish gene and protein nomenclature conventions (https://wiki.zfin.org/display/prot/Conventions+For+Naming+Zebrafish+Genes) and not use Zoxr1a to denote zebrafish gene.
  2. Provide details of effect of 8bp del on the protein sequence in Figure 3D by using the recommended mutant protein designation, e.g. p.A100Cfs*8 (hypothetical example). The authors can describe it for the longest isoform.
  3. It is not clear if the mutant and WT fish analyzed for phenotypes described in figures 4 and 6 are clutch-mates or grown from independent crosses. In figure 5, the authors mention age-matched females were used. Ideally, phenotype analysis should be done from het to het crosses and letting the fish of different genotypes grow together under the same conditions until time of phenotype analysis. The authors should describe the crosses used to generate fish that were analyzed.
  4. Authors describe lack of spawning in 5 month old mutant fish as premature ovarian failure. The authors should mention if the mutant females spawned before 5 months of age or they were not tested prior to this age. Normally fish reach sexual maturity at ~3 months. In Discussion, lines 470-471, the authors state that most mutants became infertile after 5 months of age. However, in results only females are described as infertile whereas males are fertile and no data presented for after 5 months for either males or females.
  5. It is not clear how the dose of H2O2 was selected for induction of oxidative damage. Ideally, a dose-response curve is done by testing exposure to a series of doses of the chemical or a published dose is used. In the study by Lisse et al., (Scientific Reports 6:20328), 3dpf ZF embryos were treated with H2O2 and RNA-Seq was performed. The authors should discuss their findings with relation to this paper at least in WT fish.
  6. In the model depicted in figure 12 D, it is not clear how the authors link apoptosis to premature ovarian failure and aging. The only apoptosis observed after H2O2 treatment was in olfactory bulbs.
  7. It is not clear how the authors included “antioxidant defenses” in the conclusion of expression data mentioned in lines 211-213.
  8. It is not clear what is loaded in the 2nd from right lane on the gel shown in Figure 3C. The lanes need to be labeled better.

Minor concerns and suggestions:

  • Lines 369-370: It is not the increased “number” of early stress genes but increased “level” of early stress genes.
  • Line 458: Change “this mutant was deleted in 8bp in exon 8” to “this mutation was a deletion of 8bp in exon 8”
  • Reference # 28 cited for WISH methods does not describe WISH methods, but cites another paper, so the authors should cite the original paper that actually describes WISH method so the reader can get to method details without going through multiple publications.
  • Line 108: “genotyping were screened via..” should be changed to “genotyping was performed via..”
  • Line 118: “To assessment the impact of..” should be changed to “To assess the impact of..”
  • Line 114: 3000 rmp: is it 3000 rpm?
  • Line 285: delete “of”

Author Response

Dear Reviewer:

Thanks you for your comments and constructive suggestions for revising and improving our manuscript “Zebrafish oxr1a Knockout Reveals Its Role in Regulating Antioxidant Defenses and Aging”. We have carefully considered your comments, and have modified the manuscript accordingly. Your comments encourage us to further explore the role of oxr1a in zebrafish. If there are other aspects in the manuscript that require further clarification, kindly let us know and we would be delighted to comply with.

Thanks again for all the helpful comments you provided.

Sincerely yours,

Hao Xu, Yu Jiang, Sheng Li, Lang Xie, Yi-Xi Tao and Yun Li*

Response to Reviewer Comments

Major concerns:

Point 1: The authors should follow zebrafish gene and protein nomenclature conventions (https://wiki.zfin.org/display/prot/Conventions+For+Naming+Zebrafish+Genes) and not use Zoxr1a to denote zebrafish gene.

Response 1: Thanks for your suggestion. For ease of description, we used “Zoxr1am/m” indicates the zebrafish oxr1a homozygous mutant in our article. For this naming, we refer to another article on zebrafish gene knockout (Xia et al., 2018; doi: 10.1534/genetics.117.300574). We hope that reviewers can check it out kindly. Of course, if our naming method does not meet the publication requirements, we are very willing to revise it.

Point 2: Provide details of effect of 8bp del on the protein sequence in Figure 3D by using the recommended mutant protein designation, e.g. p.A100Cfs*8 (hypothetical example). The authors can describe it for the longest isoform.

Response 2: Following the suggestions of reviewers, we have used the longest isoform (oxr1ax1) of zebrafish oxr1a gene and described the effect of 8bp deletion in exon6 on the protein sequence in Figure 3D.

Point 3: It is not clear if the mutant and WT fish analyzed for phenotypes described in figures 4 and 6 are clutch-mates or grown from independent crosses. In figure 5, the authors mention age-matched females were used. Ideally, phenotype analysis should be done from het to het crosses and letting the fish of different genotypes grow together under the same conditions until time of phenotype analysis. The authors should describe the crosses used to generate fish that were analyzed.

Response 3: We do agree with reviewer’s views. For the adult fish experiment, we obtained WT and mutant zebrafish by crossing heterozygotes and heterozygotes, ensuring that they grow and develop under the same conditions. For the larval fish experiment, because the individuals are too small to identify different genotypes by fin-cutting sequencing, we therefore obtained WT and mutants by independent mating. Following the suggestions of reviewers, we have added our description to the Result 3.3 of this article.

Point 4: Authors describe lack of spawning in 5 month old mutant fish as premature ovarian failure. The authors should mention if the mutant females spawned before 5 months of age or they were not tested prior to this age. Normally fish reach sexual maturity at ~3 months. In Discussion, lines 470-471, the authors state that most mutants became infertile after 5 months of age. However, in results only females are described as infertile whereas males are fertile and no data presented for after 5 months for either males or females.

Response 4: Thanks for your professional question. Yes, the mutant females are fertile before 3 months of age. According to our data, the fecundity of mutant females gradually declines with age. After 5 months of age, we observed that most mutant females became infertile. Through histological analyses, we found that most oocytes were arrested in the early development stages of the primary growth stage (PG), previtellogenic stage (PV) and early vitellogenic stage (EV) in ovaries. However, interestingly, the oxr1a mutation does not seems to have an effect on the fecundity of male fish. No evident differences were observed in terms of the morphology and GSI of the testis between Zoxr1am/m and WT zebrafish males. Following the suggestions of reviewers, we have added a supplementary data involving in the ovulation rate of mutant females before 5 months of age in Figure S2.

Point 5: It is not clear how the dose of H2O2 was selected for induction of oxidative damage. Ideally, a dose-response curve is done by testing exposure to a series of doses of the chemical or a published dose is used. In the study by Lisse et al., (Scientific Reports 6:20328), 3dpf ZF embryos were treated with H2O2 and RNA-Seq was performed. The authors should discuss their findings with relation to this paper at least in WT fish.

Response 5: Thanks for your suggestion, this literature (Lisse et al., 2016; doi: 10.1038/srep20328.) provides us with important data support, we therefore have added it as a reference # 33 in our article. In their study, zebrafish larvae at 3 dpf were treated with 3 mM H2O2. In fact, we have a preliminary experiment before the formal experiment. Regarding the induction of oxidative damage, the concentration of H2O2 is suitable in the range of 1 ~ 3 mM. Under 2 mM H2O2 stress, we can control the observation and recording of larval death time to about 12 hours. Therefore, this dose of 2 mM H2O2 is beneficial to assess the impact of zebrafish oxr1a knockout on the resistance to oxidative stress.

Point 6: In the model depicted in figure 12 D, it is not clear how the authors link apoptosis to premature ovarian failure and aging. The only apoptosis observed after H2O2 treatment was in olfactory bulbs.

Response 6: We do agree with reviewer’s views. It is well-known that ageing is often connected to inflammation. In our work, under H2O2 stress, we observed that an inflammatory response in Zoxr1am/m was strongly induced in larval stage, thus oxr1a may play an important role in regulating the inflammatory reaction. However, we did not provide direct evidence to support the link apoptosis to premature ovarian failure and aging in adult fish, which is our next research work. Therefore, our statement is not rigorous in the model depicted in figure 12D. Following the suggestions of reviewers, we have corrected our description in figure 12D.

Point 7: It is not clear how the authors included “antioxidant defenses” in the conclusion of expression data mentioned in lines 211-213.

Response 7: Thanks for your suggestion. According to previously reports in mice (Oliver et al., 2011; doi: 10.1371/journal.pgen.1002338) and Drosophila (Wang et al., 2019; doi: 10.1016/j.ajhg.2019.11.002), oxr1 is highly expressed in the central nervous system and participates in the antioxidant protection of neurons. In our study, a similar high expression in brain was observed in zebrafish larvae, we therefore think that oxr1a may be essential for protection against oxidative stress in the central nervous system of zebrafish. But we do agree with reviewer’s views, our description is not rigorous in lines 211-213. According to the suggestion, we have corrected “The high expression levels of oxr1a in embryos, gonads and the central nervous system suggested its important roles in early development, reproduction and antioxidant defenses” as “The high expression levels of oxr1a in embryos, gonads and the central nervous system suggested it may plays an important role in early development, reproduction and antioxidant defenses”

Point 8: It is not clear what is loaded in the 2nd from right lane on the gel shown in Figure 3C. The lanes need to be labeled better.

Response 8: The 2nd from right lane on the gel shown in Figure 3C is heterozygous mutant. As suggested by the reviewer, we have labelled the lanes again.

Minor concerns and suggestions:

Point 1: Lines 369-370: It is not the increased “number” of early stress genes but increased “level” of early stress genes.

Response 1: Following the suggestions of reviewers, we have corrected “number” as “level”.

Point 2: Line 458: Change “this mutant was deleted in 8bp in exon 8” to “this mutation was a deletion of 8bp in exon 8”

Response 2: “this mutant was deleted in 8bp in exon 8” has been corrected as “this mutation was a deletion of 8bp in exon 8”.

Point 3: Reference # 28 cited for WISH methods does not describe WISH methods, but cites another paper, so the authors should cite the original paper that actually describes WISH method so the reader can get to method details without going through multiple publications.

Response 3: Thanks for your suggestion. We have cited the original article (Maures et al., 2002; DOI: 10.1210/en.143.7.2722) to describe the WISH method in our article.

Point 4: Line 108: “genotyping were screened via..” should be changed to “genotyping was performed via..”

Response 4: We have corrected “genotyping were screened via..” as “genotyping was performed via..”.

Point 5: Line 118: “To assessment the impact of..” should be changed to “To assess the impact of..”

Response 5: We revised it as suggested.

Point 6: Line 114: 3000 rmp: is it 3000 rpm?

Response 6:  Yes, we have corrected “3000 rmp” as “3000 rpm”.

Point 7: Line 285: delete “of”

Response 7: Thank you. We revised it as suggested.

Reviewer 2 Report

Hao Xu and collegeaus present with “Zebrafish…” a comprehensive introduction of a new model system to study the function of the gene oxr1a. From previous studies in other animal models, patient data and in vitro studies it was known that oxr1a might be involved in oxidative stress response. OS is one of the important drivers of diseases or ageing and needs detailed investigations. To study this in zebrafish a crispr knock-out was generated and characterized phenotypically as well as its influence on transcriptional regulation.

In principle the report is described in good detail. Also the used techniques are chosen sensibly and were conducted extensively. However, for example in Figure 2 I would like to know n numbers and positive fish per experiment for in situ experiments.

Figure 1 describes the zebrafish oxr1a gene in good detail, however, I am missing an explanation why oxr1b is not supposed to play a role. (What might be, but it is not automatically the case that the a form is the functional form).

Figure 4 describes the influence of the knock out on life span and physiology. I find the fat assay interesting, however, ageing is often connected to inflammation related cachexia, would it be possible to also determine the protein content of the fish? Or muscle density?

Figure 6: why is apoptosis seen especially in the olfactory bulbs? Does down regulation of these certain antioxidant genes direct into a certain network of reaction towards a certain ROS molecule?

Figure 7: please specify the n number of used larvae. Here it would have been good to perform complementary experiments, if overexpression of human or at least zebrafish oxr1a RNA would rescue the observed phenotype.

Fig. 8. To fig. 12 : The analysis of the different expression patterns between wildtype and zoxr1a-/- was conducted in detail and several interpretations are discussed. I fully agree to see that p53 pathway is coming up, however, I wonder, if this is direct effect or because the antioxidant response is missing, an increase in DNA damage is generated, which leads to the p53 upregulation. Can this be detected, would it be possible to check for DNA damage?

Taken together, please try to add some literature based explanations at least for these points and please try to provide n numbers of used larvae! Thank you for undertaking the effort to provide an oxidative stress model in Zebrafish for future studies in this important field!

Author Response

Dear Reviewer,

      We really appreciate all the valuable comments you have provided, which are very helpful for revising and improving our manuscript “Zebrafish oxr1a Knockout Reveals Its Role in Regulating Antioxidant Defenses and Aging”. We have considered the comments carefully and have made revision which marked in red in the revised paper. We have tried our best to revise our manuscript according to the comments. The points raised by you have been dealt with in the revised manuscript, as detailed in the Author's Notes to Reviewer. We would like to take this opportunity to express our gratitude to you in helping us to clarify a few points in our revised manuscript. Your constructive comments have enabled us to arrive at an improved manuscript. If there are other aspects in the manuscript that require further clarification, kindly let us know and we would be delighted to comply with.

Thanks again for all the helpful comments you provided.

Sincerely yours,

Hao Xu, Yu Jiang, Sheng Li, Lang Xie, Yi-Xi Tao and Yun Li*

Response to Comments

Point 1: In principle the report is described in good detail. Also the used techniques are chosen sensibly and were conducted extensively. However, for example in Figure 2 I would like to know n numbers and positive fish per experiment for in situ experiments.

Response 1: Thanks for your comment. For whole-mount in situ hybridization, twenty larvae were pooled as one experimental sample. In our study, the number of positive fish per experiment was about 90%. As suggested by the reviewer, we have added this information to Section 3.1 of our article.

Point 2: Figure 1 describes the zebrafish oxr1a gene in good detail, however, I am missing an explanation why oxr1b is not supposed to play a role. (What might be, but it is not automatically the case that the a form is the functional form).

Response 2: Thanks for your suggestion. According to the synteny analysis, oxr1b is a duplicate gene of oxr1a on Chr19. In fact, we have studied the function of oxr1b in zebrafish. However, we found that oxr1b mainly plays an important role in early embryonic development, especially in the transition from maternal to zygotic stage. Considering that the difference in functionality, we therefore did not mention it in this article, but I will describe oxr1b in detail in our next article and compare it with oxr1a in gene structure, expression and function. Of course, if the reviewer thinks it is necessary to describe oxr1b in this article, I hope to get your further guidance.

Point 3: Figure 4 describes the influence of the knock out on life span and physiology. I find the fat assay interesting, however, ageing is often connected to inflammation related cachexia, would it be possible to also determine the protein content of the fish? Or muscle density?

Response 3: We do agree with reviewer’s views. Aging is often connected to inflammation. In fact, under H2O2 stress, we also found that the fold changes in the expression of several inflammation-related genes such as il1b (interleukin, beta), il8 and il11a in Zoxr1am/m were greater than that of WT zebrafish. Meanwhile, the Gene Ontology (GO) enrichment analysis showed that the inflammatory response in Zoxr1am/m was strongly induced, thus oxr1a may play an important role in regulating the inflammatory reaction. These findings are described in detail in our article. Regarding the fat assay in our study, we found that mutants became thinner and thinner with age, which attracted us to study the changes in total lipid content of zebrafish. This phenotype is similar to that of mice lacking OXR1 (Oliver et al., 2011; doi: 10.1371/journal.pgen.1002338), which displayed reduced food intake and weight loss, and do not survive beyond sexual maturity. Therefore, we think that the decrease in total fat content of mutants may be related to aging reducing food intake, but we need to make further observations. Regarding the protein content and muscle density of the fish, we are very sorry that we have not tested in time. But we believe that the two indicators are very important for the description of the mutant aging process, and we will test them in time in our next research.

Point 4: Figure 6: why is apoptosis seen especially in the olfactory bulbs? Does down regulation of these certain antioxidant genes direct into a certain network of reaction towards a certain ROS molecule?

Response 4: Thanks for your professional question. Olfactory bulbs are a key part of the neural system responsible for the sense of smell, which receives and processes the information from the olfactory sensory neurons in the nasal mucosa and sends it to different parts of the primary olfactory cortex in the forebrain, such as anterior olfactory nucleus, indusium griseum, taenia tecta, olfactory tubercle, piriform cortex, periamygdaloid cortex, and entorhinal cortex. In zebrafish, olfactory bulbs are located at the foremost part of the brain and is in close contact with the environment, so it is vulnerable to environmental stress. In our study, whole mount in situ hybridization showed that oxr1a is expressed in the olfactory bulb after 48 hpf (Fig. 2C), suggesting it may plays an important role in the olfactory bulbs. By TUNEL staining, we found that a high aggregation of apoptotic cells was observed in olfactory bulbs of Zoxr1am/m, but only a few TUNEL-positive cells in the WT larvae. Therefore, we can basically confirm that oxr1a has an antioxidant effect on the olfactory bulbs.

     By qPCR, we detected several antioxidant genes including gpx1b, gpx4a, gpx7 and sod3a in Zoxr1am/m which were significantly down-regulated compared to WT zebrafish (Figure 6B). These antioxidant genes are directly involved in the process of detoxifying cellular reactive oxygen species. By comparative transcriptome analysis of early oxidative stress response, we found that oxr1a mutation caused more extensive changes in transcriptional networks compared to WT zebrafish, including several stress responses, pro-inflammatory reaction, p53 signaling and apoptosis pathway. Therefore, it seems undeniable that this is probably related to the down-regulation of these certain antioxidant genes in the mutant. Regarding the second question from the reviewer, we think is very interesting, but unfortunately, we did not conduct a more in-depth study, we hope to study this content in the next work.

Point 5: Figure 7: please specify the n number of used larvae. Here it would have been good to perform complementary experiments, if overexpression of human or at least zebrafish oxr1a RNA would rescue the observed phenotype.

Response 5: Thanks for your suggestion. For whole-mount in situ hybridization, twenty larvae were pooled as one experimental sample in this study. As suggested by the reviewer, we have added this information in our article. Regarding the rescue experiment, I’m very sorry that I have not completed it in time. An important reason is that the fertility of mutants is too weak and the adult fish have a short life span. As a result, it is difficult to obtain enough embryos for microinjection experiments. But we do agree with reviewer’s views. For the scientific rigor of the experiment, we plan to raise the amount of mutants to overcome the current problem of fewer embryos, and perform the supplementary experiment as soon as possible.

Point 6: Fig. 8. To fig. 12: The analysis of the different expression patterns between wild type and Zoxr1a-/- was conducted in detail and several interpretations are discussed. I fully agree to see that p53 pathway is coming up, however, I wonder, if this is direct effect or because the antioxidant response is missing, an increase in DNA damage is generated, which leads to the p53 upregulation. Can this be detected, would it be possible to check for DNA damage?

Response 6: Thanks for your professional question. According to the RNA-seq data, the expression level of p53 in mutants were up-regulated by 55.15% compared to the wild type. However, in our study, the differential expression levels with a more than two folds change were considered as differentially expressed gene, thereby leading to this result cannot be well highlighted. Regarding whether the activation of the p53 pathway is caused by DNA damage, previously report have confirmed that oxr1 plays an important role in maintaining mitochondrial DNA integrity and avoiding apoptosis (Yang et al., 2014; DOI: 10.1016/j.freeradbiomed.2014.09.003). Therefore, combined with our research, we have reason to believe that the activation of p53 pathway and apoptosis in mutants may be caused by increased DNA damage. As suggested by the reviewer, in the next work, we intend to study the effect of zebrafish oxr1a mutation on the mitochondrial DNA integrity by the long range PCR method and the TaqI-based qPCR method as previously described (Rothfuss et al., 2009; doi:10.1093/nar/gkp1082).
